# Investigation of Monolithic 3D Integrated Circuit Inverter with Feedback Field Effect Transistors Using TCAD Simulation

**DOI:** 10.3390/mi11090852

**Published:** 2020-09-13

**Authors:** Jong Hyeok Oh, Yun Seop Yu

**Affiliations:** Department of ICT Robot Eng, Hankyong National University, 327 Jungang-ro, Anseong-si, Gyenggi-do 17579, Korea; rnjsdlr7@hknu.ac.kr

**Keywords:** monolithic 3-D integrated inverter, feedback field effect transistor, voltage transfer characteristics, transient response

## Abstract

The optimal structure and process for the feedback field-effect transistor (FBFET) to operate as a logic device are investigated by using a technology computer-aided design mixed-mode simulator. To minimize the memory window of the FBFET, the channel length (*L_ch_*), thickness of silicon body (*T_si_*), and doping concentration (*N_ch_*) of the channel region below the gate are adjusted. As a result, the memory window increases as *L_ch_* and *T_si_* increase, and the memory window is minimum when *N_ch_* is approximately 9 × 10^19^ cm^−3^. The electrical coupling between the top and bottom tiers of a monolithic 3-dimensional inverter (M3DINV) consisting of an n-type FBFET located at the top tier and a p-type FBFET located at the bottom tier is also investigated. In the M3DINV, we investigate variation of switching voltage with respect to voltage transfer characteristics (VTC), with different thickness values of interlayer dielectrics (*T_ILD_*), *T_si_*, *L_ch_*, and *N_ch_*. The variation of propagation delay of the M3DINV with different *T_ILD_*, *T_si_*, *L_ch_*, and *N_ch_* is also investigated. As a result, the electrical coupling between the stacked FBFETs by *T_ILD_* can be neglected. The switching voltage gaps increase as *L_ch_* and *T_si_* increase and decrease, respectively. Furthermore, the slopes of VTC of M3DINV increase as *T_si_* and *N_ch_* increase. For transient response, *t_pHL_* decrease as *L_ch_*, *T_si_*, and *N_ch_* increase, but *t_pLH_* increase as *L_ch_* and *T_si_* increase and it is almost the same for *N_ch_*.

## 1. Introduction

According to Moore’s Law, the number of transistors in integrated circuit (IC) chips is increasing with the improvements in semiconductor performance [1,2]. To improve the performance of next generation semiconductors, the requirement of integrating more transistors in a smaller area using nanoscaling is increasing. However, semiconductor devices with in scaling below 10 nm have fabrication challenges [3,4]. To overcome these challenges, monolithic 3-dimensional ICs (M3DICs) are attracting attention. M3DICs are designed with a transistor, a logic gate, and system blocks, which are integrated vertically to equip the M3DIC in accomplishing higher integration than a 2D conventional circuit. In ICs, each block is connected using a vertical interconnect that can be shorter than a horizontal interconnect and can achieve a lower critical delay [5,6,7,8,9,10,11]. Each vertically stacked block has an electrical coupling between the upper and lower tiers. Research on the application of monolithic 3D integration (M3DI) is being actively conducted and electrical coupling in the design of M3DI is being studied [11,12,13,14,15,16]. In particular, electrical coupling must be considered for the circuit design to achieve the desired performance.

Among the various applications of M3DI [17,18,19,20,21], the feedback field-effect transistor (FBFET) is in the spotlight as a next-generation device [22,23]. The FBFET has a S-shape energy band caused by p-n-p-n structure for n-type devices and it works on the principle of positive feedback. The positive feedback is caused by different types of doping in the channel region. Further to this, the channel region of FBFET is divided into two regions. One is the channel region below the gate and the other is channel region except for that below the gate. These two regions form an energy barrier owing to the n–p junction, and those regions form an energy barrier between the source/drain-side channel region and source/drain region, owing to the p–n junction. To operate the FBFET, the energy barrier caused by doping concentration of the source region and drain region must be removed. This process makes the energy band of the source-drain region parallel and achieves a clear saturation mode at on-state. When gate bias applied, the energy band is lower, and the electrons at the source region can flow along the drain-source electric field. Subsequently, electron stacks in the energy well formed between the channel and drain regions, and continuous stacking removed the energy well. This mechanism makes the energy band of all regions parallel and this happens in a short time [23]. According to the mechanism, FBFET has approximately zero subthreshold swing and hysteresis characteristics. Therefore, it can be used as a logic and memory device with the same structure. Various applications of FBFET were studied, such as using it as a logic device, a memory device, and a neuron circuit [24,25,26,27,28,29,30]. In the previous study, we investigated electrical coupling between vertically stacked FBFETs in the monolithic 3-dimensional inverter (M3DINV) with FBFETs, in terms of device characteristics [31]. However, studies on electrical interaction in terms of the electrical characteristics of an inverter are yet to be reported. The FBFET has a memory window, so it is necessary to investigate the changes in the electrical characteristics of the inverter. In addition, when configured with an M3DIC, an investigation of inverter characteristic variation is required for electrical coupling.

In this study, the electrical characteristics of an M3DINV consisting of vertically stacked FBFETs (M3DINV-FBFET) using a technology computer-aided design (TCAD) mixed-mode simulator are analyzed. The simulation structure parameters of M3DINV-FBFET are described in Section 2. In Section 3, the mechanism of FBFET through the energy bands is described. Moreover, the structure which memory windows of top N-type and bottom P-type FBFETs minimize is investigated with variation of its channel lengths, thickness of silicon body, and doping concentrations of the channel region. In addition, the extent to which the switching voltages of the voltage transfer characteristics (VTCs) and propagation delay of M3DINV-FBFET depend on the thickness of interlayer dielectrics (ILD) is described. Section 4 presents the conclusion of this study.

## 2. Simulation Structure Parameters

Figure 1a shows the 3D schematics of the M3DINV-FBFET and Figure 1b shows a-a’ cross-sectional views of the M3DINV-FBFET. The structure parameters are summarized in Table 1. Atlas [32] of Silvaco was used for the TCAD simulation. The 2D structure was used for simulation, as shown in Figure 1b. To simulate FBFET accurately, the models of bipolar junction transistor (BJT) and metal–oxide–semiconductor field-effect transistor (MOSFET), including the Lombardi concentration, voltage, and temperature (CVT) model, Shockley–Read–Hall model, Fermi–Dirac for MOSFET and field-dependent mobility model, Auger recombination model, and band gap narrowing model for BJT, are used. The simulation temperature was 300 K.

As shown in Figure 1b, the p-type and n-type FBFETs are located at Tier1 and Tier2, respectively. Gate-oxide was composed of hafnium dioxide (HfO_2_). The length of the gate was half the length of *L_ch_* and was located on the source-side in the channel region. The FBFET had a p-n-p-n/n-p-n-p (n-type/p-type) structure. Uniform doping concentrations in the n-type and p-type FBFETs were adopted in each region. The gate work functions of the p-type and n-type FBFETs were 4.8 and 4.6 eV, respectively.

## 3. Simulation Results

Figure 2 shows the energy band diagrams of n-type FBFET. The red and black lines denote conduction and valence bands, respectively. The symbols ‘e’ and’ ‘h’ represent electron and hole, respectively. To operate as FBFET, first, the energy barrier owing to the p–n junction between the drain–source region must be removed. As the drain–source voltage (*V_ds_*) applies to FBFET, the energy band of the drain and source regions is aligned, as shown in Figure 2a. When forward sweep is proceeded to FBFET, the energy band below the gate is lower. The electrons at the source region flow along the electric field and trap well in the energy, as shown in Figure 2b. The continuous stacking electrons remove the energy well. As a result, the energy band of all regions is aligned, as shown in Figure 2c. Moreover, all carriers flow freely along the electric field. By contrary, when reverse sweep is proceeded to FBFET at on-state, the energy band below the gate is higher and the energy barrier blocks the flow of carriers. As shown in Figure 2d, the energy barrier and well form owing to accumulating carriers.

Figure 3 shows the drain current–gate voltage (*I_ds_* − *V_gs_*) characteristics and the drain current–drain voltage (*I_ds_* − *V_ds_*) characteristics of the top n-type and the bottom p-type FBFETs in the M3DINV-FBFET. An arrow pointing up denotes forward sweep, and pointing down denotes reverse sweep. FBFET works on a positive feedback when the carriers flow [22]. This mechanism causes a steep increase in current, as shown in Figure 3a. The difference in the threshold voltage between the forward and reverse sweeps is called the memory window. The memory window of p-type FBFET was 0.09 V at *V_ds_* = −1 V, and that of the n-type FBFET was 0.21 V at *V_ds_* = 1 V and *V_pgs_* = 0 V, respectively.

### 3.1. Memory Window

For adjusting the memory window of the M3DINV-FBFET, various structure parameters of FBFET can be changed. In this study, the memory windows of the top and bottom FBFETs in the M3DINV-FBFET were investigated with the structure parameters *T_si_*, *L_ch_*, and *N_ch_*. Figure 4a shows the variation of memory window at different channel lengths. Channel length was changed from 80 to 160 nm. The memory windows of both types of FBFETs became smaller as *L_ch,_* was increased, as shown in Figure 4a. Memory window of the n-type FBFET was decreased from 0.21 V at 80 nm to 0.11 V at 160 nm and of the p-type, FBFET was decreased minutely from 0.09 V at 80 nm to 0.08 V at 160 nm. In addition, as *L_ch_* was increased, the threshold voltages of both types of FBFET were shifted in an arrow-direction, as shown in Figure 4a. To operate a device as logical, it must be minimized to approximately zero. However, the FBFET has a hysteresis with the memory window. To minimize the memory window, *L*_ch_ for n-type was increased and that of p-type FBFETs was ~80 nm. Figure 4b shows the variation of memory window of the M3DINV-FBFET with *T_ILD_* = 100 nm and *L_ch_* = 80 nm at different *T_si_* (from 4 to 30 nm). The memory window of both types of FBFETs became smaller as *T_si_* was increased, and the on-current of n-type FBFET was increased about ten times, as shown in Figure 4b. The memory window of n-type FBFET was decreased from 0.23 V at 4 nm to 0.14 V at 30 nm and that of p-type FBFET minutely was decreased from 0.09 V at 4 nm to 0.08 V at 30 nm. The threshold voltage of n-type FBFET was shifted in an arrow-direction, especially reverse sweep; however, the threshold voltage of p-type FBFET slight was shifted, as shown in Figure 4b. To minimize the memory window, *T_si_* was increased for both type of FBFETs.

Figure 5 shows the variation of memory window according to *N_ch_*. The *N_ch_* was changed from 4 × 10^19^ to 4 × 10^20^ cm^−3^, and the structure parameter in Table 1, except for *N_ch_*, was used for the simulation. The black and red lines denote the memory windows of n-type and p-type FBFETs, respectively. The minimum memory window for n-type FBFET was 0.20 V at 9 × 10^19^ cm^−3^. The memory window for p-type FBFET was 0.05 V at 4 × 10^19^ cm^−3^, and that was increased to 0.32 V at 4 × 10^20^ cm^−3^, as the *N_ch_* was increased. To utilize FBFET for a logic device, the minimization of memory window is required, and thus, in this study, 1 × 10^20^ cm^−3^ was used for both types of FBFETs at the channel region below the gate.

### 3.2. Voltage Transfer Characteristics

The VTC of the M3DINV-FBFET had a switching voltage gap between the forward and reverse sweeps. This occurs owing to the hysteresis characteristics of FBFET. In addition, the currents of M3DINV-FBFETs did not reach the clear on/off states. When the input voltage applied was 0 and 1 V, the output voltages were approximately 0.85 and 0.15 V, respectively. Switching voltage is affected by the electrical coupling between the top and bottom FBFETs and the window memory of FBFETs. Figure 6 shows VTC with different values of *T_ILD_*. The inset in Figure 6 shows the magnification near switching voltage of the output. As the switching voltage gaps at *T_ILD_* = 10 and 100 nm were approximately 12 and 10 mV, respectively, the change in the switching voltage gap because of *T_ILD_* was miniscule. In this structure of the M3DINV-FBFET, the electrical coupling between the stacked FBFETs with respect to *T_ILD_* can be neglected.

Figure 7 shows the VTC with different values of *L_ch_*, *T_si_*, and *N_ch_*. As explained in Section 3.1, the memory window was decreased as *L_ch_* was increased; however, the switching voltage gap in the M3DINV-FBFET was increased, as shown in Figure 7a. The inset in Figure 7a shows the magnification near switching voltage of the output. The switching voltage gaps at *L_ch_* = 80 and 160 nm were approximately 10 and 50 mV, respectively. The variation of VTC of the M3DINV-FBFET with different *T_si_* is shown in Figure 7b. The memory window was decreased as *T_si_* was increased, as explained in Section 3.1; however, the switching voltage gap and slope of the M3DINV-FBFET was increased and decreased a little, respectively, as shown in Figure 7b. The variation of VTC of the M3DINV-FBFET with different values of *N_ch_* is shown in Figure 7c. In the simulation, the values of *N_ch_* of top n-type and bottom p-type FBFETs are the same with different types. The slope of the M3DINV-FBFET was increased as the *N_ch_* was increased, and the VTC for the inverter disappeared at the doping concentration of below 4 × 10^19^ cm^−^^3^. For better VTC of the M3DINV-FBFET, the doping concentration of over 9 × 10^19^ cm^−^^3^ is required. As shown in Figure 7, the switching voltage gap significantly is affected by *L_ch_* and *N_ch_*, because the threshold voltage of forward and reverse sweep is changed according to variation of *L_ch_* and *N_ch_*. Meanwhile, the switching voltage gaps with different values of *T_si_* is almost the same, because the threshold voltage is changed minutely at forward sweep and reverse sweep.

### 3.3. Transient Response

Figure 8 shows the transient simulation results of the M3DINV-FBFET at time steps of 1 μs, 1 ms, and 1 s. For simulating the transient response of the M3DINV-FBFET, a time step of at least 1 ms is required for desirable inverter characteristics. The structure parameter of the M3DINV-FBFET in Table 1 was used for simulation.

Figure 9 shows the transient response results of the M3DINV-FBFET with different values of *T_ILD_*, *L_ch_*, *T_si_*, and *N_ch_*. The insets in Figure 9 show the magnification near ‘0.5 V’ of output. The black line denotes clock and the other lines are output. The time step used is 1 s, which shows the desirable inverter characteristics among the results shown in Figure 8. High to low delay (*t_pHL_*) and low to high delay (*t_pLH_*) of the M3DINV-FBFET with the parameters in Table 1 are 28 and 46 ms, respectively. As *T_ILD_* was decreased from 100 to 10 nm, *t_pHL_* was increased from 28 to 29 ms, and *t_pLH_* was almost the same, as shown in Figure 9a. As *L_ch_* was increased from 80 to 160 nm, *t_pHL_* was decreased from 29 to 6 ms; however, *t_pLH_* was increased from 46 to 70 ms, as shown in Figure 9b. As *T_si_* was increased from 4 to 30 nm, *t_pHL_* decreased from 29 to 24 ms, whereas *t_pLH_* was increased 44 to 49 ms, as shown in Figure 9c. As *N_ch_* increases from 4 × 10^19^ to 1 × 10^20^ cm^−^^3^, *t_pHL_* decreased from 53 to 28 ms, whereas *t_pLH_* was almost the same, as shown in Figure 9d. The variation of *t_pHL_* and *t_pLH_* by *T_ILD_* was very small and thus, the electrical coupling between the stacked FBFETs with respect to *T_ILD_* can be neglected. As all *L_ch_*, *T_si_*, and *N_ch_* increased, all the *t_pHL_* decreased, whereas the *t_pLH_* increased, increased, and was almost the same, respectively.

## 4. Conclusions

In this study, we investigated the optimum structure of top n-type and bottom p-type FBFETs in an M3DINV-FBFET to minimize the memory window of each FBFET using TCAD mixed-mode simulation. Furthermore, the electrical coupling between the top and bottom tiers in the M3DINV-FBFET was studied. First, we investigated the memory window with the various values of *L*_ch_, *T_si_*, and *N_ch_* of FBFETs in the M3DINV-FBFET. It was observed that as all the *L_ch_* and *T_si_* increase, all the memory windows decreased. The minimum memory window for n-type FBFET is 0.20 V at 9 × 10^19^ cm^−^^3^, and that of p-type FBFET was increased as *N_ch_* increased. In the M3DINV-FBFET, the VTC had a switching voltage gap owing to the hysteresis characteristics of FBFET. The variation of switching voltage gap by *T_ILD_* was very small (~10 mV), and thus, the electrical coupling between the stacked FBFETs with respect to *T_ILD_* can be neglected. The switching voltage gap increased as *L_ch_* and *T_si_* increased and decreased, respectively. Furthermore, the slope of VTC of the M3DINV-FBFET increased, as the *T_si_* and *N_ch_* increased. The transient response of the M3DINV-FBFET was investigated with respect to *T_ILD_*, *L_ch_*, *T_si_*, and *N_ch_*. The variation of *t_pHL_* and *t_pLH_* by *T_ILD_* was very small and thus, the electrical coupling between the stacked FBFETs with respect to *T_ILD_* can be neglected. As all *L_ch_*, *T_si_*, and *N_ch_* increased, all the *t_pHL_* decreased, whereas the *t_pLH_* increased, increased, and was almost the same, respectively. Even though conclusively, the M3DINV-FBFET may be tedious to use in logic circuits due to ambiguous on/off states and the relatively slow time response (approximately several decade ms), it is necessary to further investigate the optimal structure and process for the FBFET to operate as a logic device.

## Figures and Tables

**Figure 1 micromachines-11-00852-f001:**
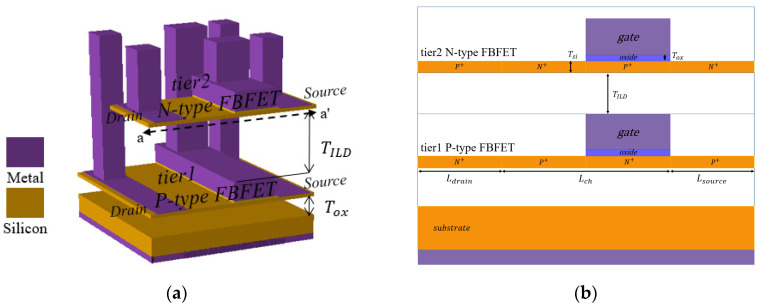
(**a**) 3D schematic of M3DINV-FBFET; (**b**) a-a′ cross-sectional view of M3DINV-FBFET.

**Figure 2 micromachines-11-00852-f002:**
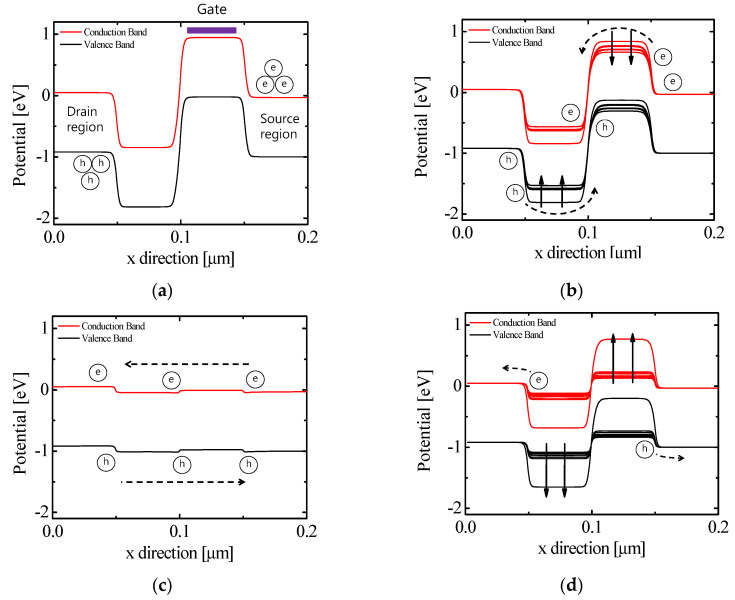
Energy band diagram of n-type FBFET at *V_ds_* = 1 V. (**a**) *V_gs_* = 0 V; (**b**) forward sweep, *V_gs_* = 0 to 1 V; (**c**) *V_gs_* = 1 V; and (**d**) reverse sweep, *V_gs_* = 1 to 0 V.

**Figure 3 micromachines-11-00852-f003:**
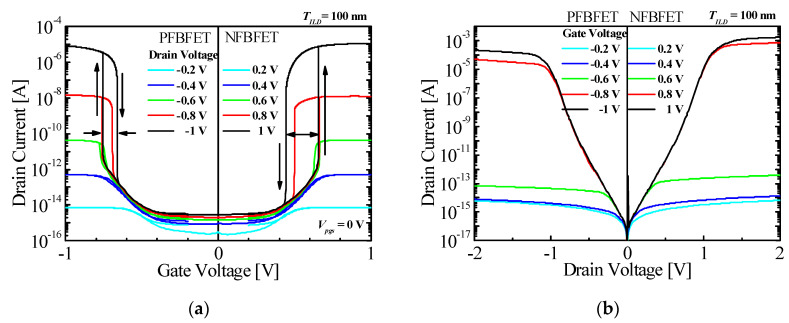
Transfer characteristics of P-type and N-type FBFETs. (**a**) *I_ds_−V_gs_* characteristics and (**b**) *I_ds_−V_ds_* characteristics.

**Figure 4 micromachines-11-00852-f004:**
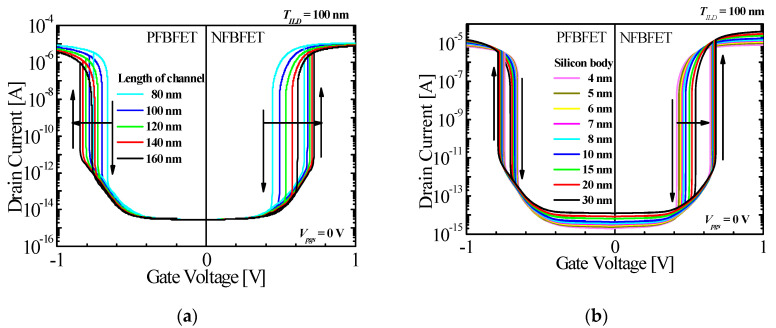
*I_ds_* − *V_gs_* characteristics of P-type and N-type FBFETs in M3DINV with different values of (**a**) *L_ch_* and (**b**) *T_si_* (*T_ILD_* = 100 nm, for N-type FBFET, *V_pgs_* = 0 V).

**Figure 5 micromachines-11-00852-f005:**
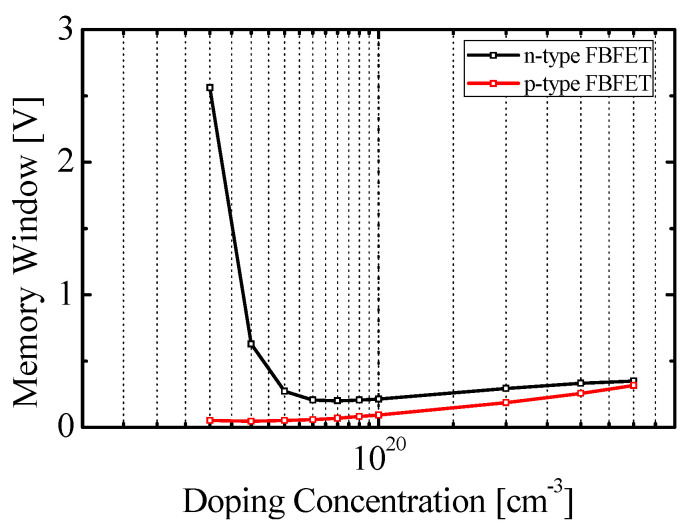
The memory window with different value of *N_ch_*.

**Figure 6 micromachines-11-00852-f006:**
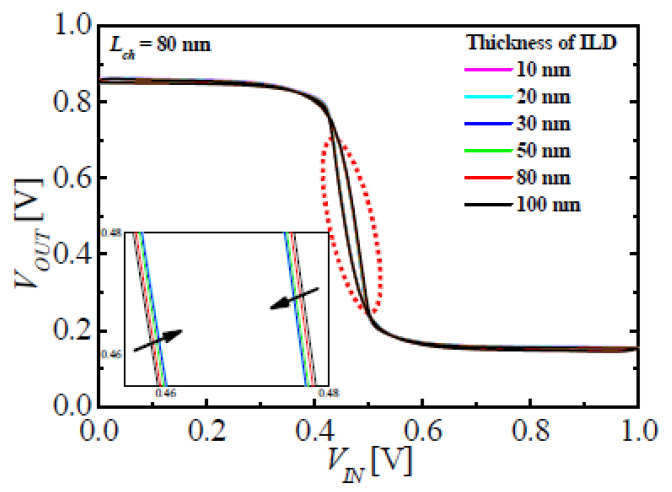
Voltage transfer characteristics (VTC) of M3DINV-FBFET with *L_ch_* = 80 nm at different values of *T_ILD_*.

**Figure 7 micromachines-11-00852-f007:**
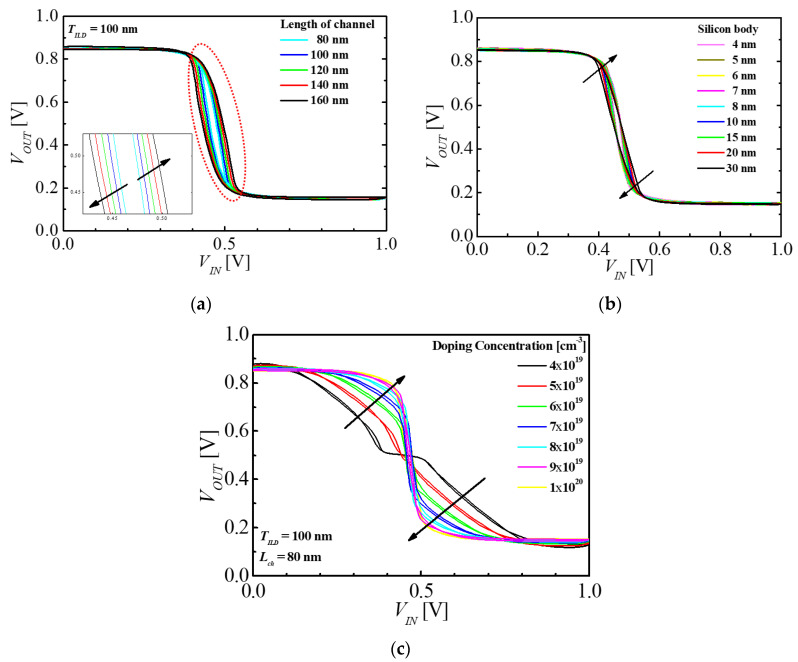
VTC of M3DINV-FBFET with *T_ILD_* = 100 nm at different values of (**a**) *L_ch_*, (**b**) *T_s_*_i_, and (**c**) *N_ch_*.

**Figure 8 micromachines-11-00852-f008:**
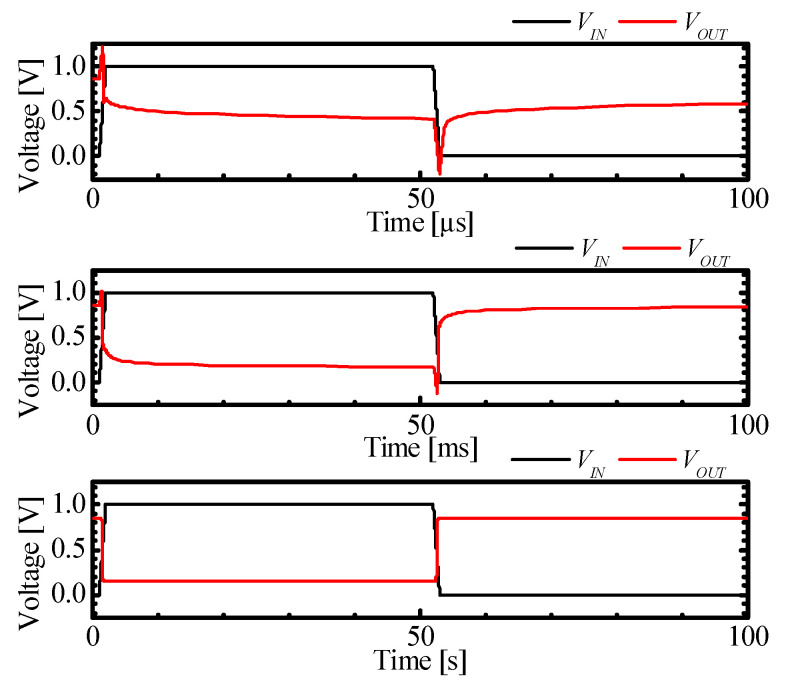
Transient response with 1 μs, 1 ms, and 1 s time steps.

**Figure 9 micromachines-11-00852-f009:**
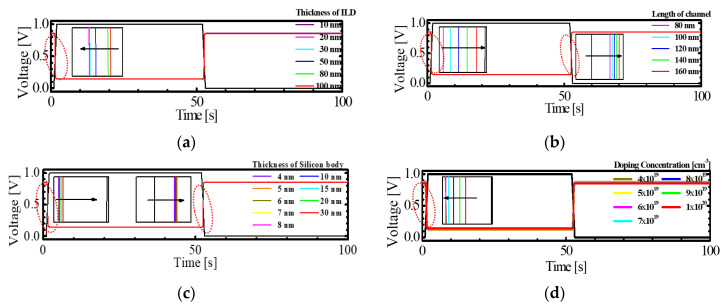
Transient response simulation results with different values of (**a**) *T_ILD_*, (**b**) *L_ch_*, (**c**) *T_si_*, and (**d**) doping concentration of channel region below the gate.

**Table 1 micromachines-11-00852-t001:** Structural parameters of M3DINV-FBFET used in technology computer-aided design (TCAD) simulation.

Parameters	Description	Value/Unit
*L_drain_*	Length of drain region	30 nm
*L_chl_*	Length of channel region	80 nm
*L_source_*	Length of source region	30 nm
*T_si_*	Thickness of silicon body	6 nm
*T_ILD_*	Thickness of interlayer dielectric	100 nm
*T_ox_*	Thickness of oxide	3 nm
*W*	Width	1 μm
*N_ch_*	Doping concentration of channel region below the gate	1 × 10^20^ cm^−3^
	P^+^ region doping concentration	1 × 10^20^ cm^−3^
	N^+^ region doping concentration	1 × 10^20^ cm^−3^
	Substrate region doping concentration	1 × 10^15^ cm^−3^
*Φ_P_*	Gate work function of P-type FBFET	4.8 eV
*Φ_N_*	Gate work function of N-type FBFET	4.6 eV

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
