# Peer review of "Investigation of Monolithic 3D Integrated Circuit Inverter with Feedback Field Effect Transistors Using TCAD Simulation"

_micromachines, 2020, doi:10.3390/mi11090852_

Round 1

Reviewer 1 Report

Monolithic 3-dimensional ICs are the new attractive trend in the integrated circuit industry, and FBFET is in the spotlight as a next-generation device. This paper proposed an optimal structure of FBFET and simulations in are presented to analyze the performances in different  channel lengths, thickness of silicon bodies, etc.. 

The paper is well written and the work has been done carefully. 

My only concern about this paper is that the contribution should be summarized more clearly, and the conclusion from each simulation can be clarified in more details in Section III. 

Author Response

Comments and Suggestions for Authors

Monolithic 3-dimensional ICs are the new attractive trend in the integrated circuit industry, and FBFET is in the spotlight as a next-generation device. This paper proposed an optimal structure of FBFET and simulations in are presented to analyze the performances in different channel lengths, thickness of silicon bodies, etc.

The paper is well written and the work has been done carefully.

-> Thanks for your comments.

My only concern about this paper is that the contribution should be summarized more clearly, and the conclusion from each simulation can be clarified in more details in Section III.

-> We added the summary of each simulation at L. 169-173, P.7 and L.191-194, P.9 in Sec. 3.

Reviewer 2 Report

  • The title should be modified to clearly show that this is a simulation study. (see for example https://www.mdpi.com/2076-3417/10/14/4977/htm)
  • Lines 13 and 22: I think the way the various increases and decreases of parameters is written in a very confusing way. I had to read the sentence several times to make some sense out of it. Is it really necessary to write all of them in the abstract in such condensed fashion?
  • Lines 48 to 53. To clearly explain the feedback mechanism, I suggest the author to add the following reference https://www.mdpi.com/2076-3417/10/9/3070/htm
  • Lines 57 to 68. The whole text is very heavy to read. The author changes very frequently from active (“we investigated”) to passive (“is investigated). Lines 66-67-68 should be rephrased to make a simpler sentence.
  • Line 75. Did you use 3D or 2D simulations?
  • Line 75. What models did you use for carrier recombination? Shockley–Read–Hall? Auger?
  • Line 75. What temperature did you set in the simulation?
  • Lines 99-101. I think this concept has already been explained in line 86. Why writing it again?
  • Caption of Figure 2 is missing the description of panels (c) and (d).
  • Line 110-112. Again difficult sentence to read/understand.
  • Line 130. Doping concentration “was” changed.
  • Line 205. Missing parenthesis

Author Response

Comments and Suggestions for Authors

The title should be modified to clearly show that this is a simulation study. (see for example https://www.mdpi.com/2076-3417/10/14/4977/htm)

-> We agree your comments. According to your suggestions, we changed the title in L. 2-4, P. 1 as follows:

Investigation of Monolithic 3D Integrated Circuit Inverter with Feedback Field Effect Transistors Using TCAD Simulation

Lines 13 and 22: I think the way the various increases and decreases of parameters is written in a very confusing way. I had to read the sentence several times to make some sense out of it. Is it really necessary to write all of them in the abstract in such condensed fashion?

-> We agree your comments. According to your suggestions, we modified a few sentences in L. 14-15, L. 23-24, P. 1.

Lines 48 to 53. To clearly explain the feedback mechanism, I suggest the author to add the following reference https://www.mdpi.com/2076-3417/10/9/3070/htm

-> We agree your comments. According to your suggestions, we added Ref. 16 in L. 55, P. 2 and the Reference.

Lines 57 to 68. The whole text is very heavy to read. The author changes very frequently from active (“we investigated”) to passive (“is investigated).

-> We agree your comments. According to your suggestions, we modified them with short text to read easily in L. 66-72, P. 2.

-> We also modified more sentences than L.57-68 as passive voice.

Lines 66-67-68 should be rephrased to make a simpler sentence.

-> We agree your comments. According to your suggestions, we modified them with simpler sentence in L. 66-69, P. 2

Line 75. Did you use 3D or 2D simulations?

-> We used 2D simulations for study. We added L.76-77, P. 2.

Line 75. What models did you use for carrier recombination? Shockley–Read–Hall? Auger?

-> The bipolar junction transistor and MOSFET models were used for FBFET simulation. We added the models in L. 77-81, P. 2.

Line 75. What temperature did you set in the simulation?

-> The simulation temperature is 300 K. We added simulation temperature in L. 80-81, P. 2.

Lines 99-101. I think this concept has already been explained in line 86. Why writing it again?

-> We agree your comments. According to your suggestions, we deleted them at L.106, P.4.

Caption of Figure 2 is missing the description of panels (c) and (d).

-> We added caption of Figure 2, as following; (c) Vgs = 1 V, and (d) reverse sweep; Vgs = 1 V to 0 V.

Line 110-112. Again difficult sentence to read/understand.

-> We agree your comments. According to your suggestions, we modified it clearly in L. 114-115, P. 5.

Line 130. Doping concentration “was” changed.

-> We corrected it in L. 133, P. 6.

Line 205. Missing parenthesis

-> We corrected it in L. 208-211, P.9.

Round 2

Reviewer 2 Report

I am satisfied with the changes